# Genetic Engineering of Starch Biosynthesis in Maize Seeds for Efficient Enzymatic Digestion of Starch during Bioethanol Production

**DOI:** 10.3390/ijms24043927

**Published:** 2023-02-15

**Authors:** Liangjie Niu, Liangwei Liu, Jinghua Zhang, Monica Scali, Wei Wang, Xiuli Hu, Xiaolin Wu

**Affiliations:** 1National Key Laboratory of Wheat and Maize Crop Science, Henan Agricultural University, Zhengzhou 450002, China; 2College of Life Sciences, Henan Agricultural University, Zhengzhou 450002, China; 3Key Laboratory of Enzyme Engineering of Agricultural Microbiology, Ministry of Agriculture and Rural Affairs, Henan Agricultural University, Zhengzhou 450002, China; 4Department of Life Sciences, University of Siena, 53100 Siena, Italy

**Keywords:** bioethanol, maize seeds, yeast fermentation, proteomic analysis, modifying starch synthesis, genetic engineering techniques

## Abstract

Maize accumulates large amounts of starch in seeds which have been used as food for human and animals. Maize starch is an importantly industrial raw material for bioethanol production. One critical step in bioethanol production is degrading starch to oligosaccharides and glucose by α-amylase and glucoamylase. This step usually requires high temperature and additional equipment, leading to an increased production cost. Currently, there remains a lack of specially designed maize cultivars with optimized starch (amylose and amylopectin) compositions for bioethanol production. We discussed the features of starch granules suitable for efficient enzymatic digestion. Thus far, great advances have been made in molecular characterization of the key proteins involved in starch metabolism in maize seeds. The review explores how these proteins affect starch metabolism pathway, especially in controlling the composition, size and features of starch. We highlight the roles of key enzymes in controlling amylose/amylopectin ratio and granules architecture. Based on current technological process of bioethanol production using maize starch, we propose that several key enzymes can be modified in abundance or activities via genetic engineering to synthesize easily degraded starch granules in maize seeds. The review provides a clue for developing special maize cultivars as raw material in the bioethanol industry.

## 1. Introduction

Biofuel, e.g., bioethanol, is the energy produced from biological sources, mainly higher plants and photosynthetic algae. The increasing use of bioethanol can reduce CO_2_ and harmful substances in automobile exhaust fumes, and alleviate the global energy crisis [1]. In the last two decades, global biofuel production increased by over 94%, because many countries are replacing a portion of their fossil fuels with bioethanol. As major producer of bioethanol, the USA and Brazil produce about 57.7 billion L and 27.6 billion L of bioethanol annually, respectively [2]. 

To date, bioethanol fermentation technology has experienced three generations of progress based on different feedstocks. The first-generation bioethanol is produced from starchy feedstocks (e.g., maize and wheat seeds) and sugar-rich feedstocks (e.g., sugarcane stalks) [3]. The second-generation bioethanol is produced from lignocellulose (e.g., maize stalks, grass) [4]. The third-generation bioethanol is produced from microalgal biomass [5]. However, second- and third-generation bioethanol have obvious drawbacks. In particular, the lignocellulosic biomass with complex and rigid structure is difficult to degrade, and ethanol productivity is relatively low [2,4]. Additionally, algae have the higher cultivation cost, plus many harvesting and extraction steps [6]. Therefore, second- and third-generation bioethanol have not yet been widely used on a large scale. The first-generation bioethanol remains the main technology for factory production of bioethanol due to its low cost.

The first-generation bioethanol needs to hydrolyze seed starch into oligosaccharides and glucose mainly via α-amylase and glucoamylase at high temperatures using a classical (pressure) cooking method [7]. That is, starch enzymatic hydrolysis in bioethanol production requires high energy inputs and additional cooking equipment, resulting in an increased cost. Alternatively, the ‘cold starch hydrolysis’ method has been developed in starch processing without liquefaction and cooking steps; thus, the energy consumption and costs can be reduced to some extent. Natural starch displays inert physicochemical features (e.g., low freeze–thaw stability, low solubility, and easy retrogradation), and it usually needs to be modified by physical (thermal and non-thermal), chemical or enzymatic methods [8,9]. Granular starch is insoluble in aqueous media at temperatures below gelatinization; thus, the amylolytic complex (e.g., endoamylases, exoamylases and debranching enzymes) and accessory hydrolases (e.g., cellulases, xylanases and proteases) need to be added for highly effective hydrolysis conversions of granular starch in the ‘cold starch hydrolysis’ method [10,11,12]. Therefore, it is necessary to develop raw starch materials that can be easily degraded for improved bioethanol production. 

As a high photosynthetic efficiency C4 plant, maize has a high biomass yield, with a great advantage in biofuel production. It accumulates a high content of starch (72% dry mass) in seeds, which is the main raw material for bioethanol production [13,14]. One bushel (25.4 kg) of maize can yield 11.0 L of ethanol, plus 8.2 kg of dry distillers’ grains via dry milling fermentation [13]. Currently, maize is the preferred raw material for the first-generation bioethanol production. With the increase in world population and the expansion of bioethanol industry, the demand for maize production will be increasing. In the USA, dent maize and tropical maize have been exclusively used for bioethanol fermentation due to their high yield and stress tolerance [15,16]. In China, about 1.43 × 10^7^ tons of cereal seeds (maize, rice and wheat), most of which were deteriorated due to improper storage, or infected by pests and diseases, are used for bioethanol production [1]. China contributes 21% of the global maize production, whereas China’s bioethanol production remains at the development stage, with a yield of 3.3 billion L in 2020 (3.4% of the world total; https://afdc.energy.gov/data/10331; accessed on 1 June 2021). 

To increase economic benefits for the starch to ethanol conversion process, it is necessary to develop specific maize for improved bioethanol fermentation. To our knowledge, there is only a report on screening maize cultivars suitable for bioethanol fermentation. Among six maize hybrids investigated in Serbia, the hybrid ZP 434 was found to be most suitable for bioethanol production, due to its higher level of soft endosperm and more easily degradation by starch-hydrolyzing enzymes [17]. Genetic engineering technology can accurately operate the target DNA sequences to obtain plants with ideal agronomic traits, but most efforts aimed to obtain crops with high yield, high quality and stress resistance [18,19,20,21,22]. Thus far, only a few maize lines with different composition, structure and properties of starch were obtained by genetic engineering techniques. For example, a high-amylose maize line was obtained using RNA interference (RNAi) in SBEIIa or SBEIIb [23,24], and a high amylopectin (waxy) was obtained by knocked out the gene *Wx1* [20]. Therefore, although some maize cultivars have been modified on starch compositions to better meet food and industrial needs, currently, there are no specific maize varieties targeted for bioethanol production.

The starch biosynthesis in maize endosperms is a complex process that requires the coordination of various enzymes and regulatory factors. It is vital to understand the key steps of starch biosynthesis pathway and screen potential targets for efficient enzymatic digestion of starch during bioethanol production. Great research advances have been made in understanding of starch synthesis pathway in cereal (e.g., maize, wheat and rice) seeds, especially by proteomic and molecular approaches. The proteome represents the total set of proteins produced by an organism or a system at a particular time or state [25]. These proteomic studies have revealed that starch biosynthesis is completed by a set of proteins in a coordinated manner in maize seeds [26]. The identified key enzymes and proteins that play the vital roles in starch synthesis can be used as potential targets for creating special maize with easily degraded starch for improved bioethanol fermentation. 

In this review, we introduced the process and characteristics of bioethanol fermentation from cereal seed starch, and discussed the factors affecting the efficiency of bioethanol fermentation. Especially, proteomic advances in identification of the key enzymes and proteins involved in maize starch biosynthesis were emphasized. It is proposed that specific enzymes can be modified by genetic engineering techniques to obtain special maize with easily degraded starch for improved bioethanol production.

## 2. Technological Process of Bioethanol Production from Cereal Seed Starch

For bioethanol production using cereal seed starch, the conversion of starch to glucose usually involves four main steps: milling, liquefaction, saccharification and yeast fermentation [14,27]. The milling of maize seeds includes wet milling and dry grind processes (Figure 1). The two processes differ in the extraction method and the resultant co-products [3].

In the wet milling process, maize kernels are steeped and fractionated into different components (e.g., starch, germ and fiber), which are separately processed to produce ethanol (from starch) and co-products (e.g., sweetener, oil and gluten meal). In the dry grind process, maize kernels are screened and cleaned to remove impurities (e.g., stones and sticks) and milled to produce ethanol along with only one co-product: distillers’ dried grains with solubles (DDGS) [13,14]. Obviously, the dry grind process requires less equipment and is more efficient in producing ethanol than the wet milling process. In the USA, dry grind ethanol production represents the majority (>70%) of ethanol processing [13]. The increasing trend in the dry-grind ethanol industry is expected to continue in the coming years due to its low cost. 

Liquefaction or dextrinization is accomplished using jet-cookers that inject steam into maize flour slurry to cook it at 90–105 °C for 1–3 h or 165 °C for 3–5 min [28,29]. After the cooked mash is cooled to 80–90 °C, thermostable α-amylase (EC 3.2.1.1) and granular starch hydrolysis enzymes (GSHE) are added in the conventional protocol (left panel, Figure 1) and the granular starch hydrolysis (GSH) protocol (right panel, Figure 1), respectively. Then, the mash is liquefied for 2 h, resulting in production of dextrins [27,30].

In the saccharification step, dextrins are degraded into glucose by adding glucoamylase (EC 3.2.1.3) or GSHE after the mash is cooled to 55–60 °C [31] or 32 °C [32]. This step takes more time than liquefaction. Glucoamylase can maintain the activity at higher temperature, making the reaction faster [33]. It is estimated that the energy costs for cooking starch represents 10–20% of bioethanol price [11,29]. 

Different from the conventional fermentation process, the ‘cold starch hydrolysis’ process, includes feedstock milling and saccharification and yeast fermentation steps, without previous high-temperature cooking and liquefaction [10] (Figure 1). This method has some disadvantages due to the omission of previous high-temperature cooking and liquefaction: (i) more hydrolyzing enzymes (e.g., pullulanase, protease and cellulase) with high efficiency need to be added; (ii) more chemicals and antibiotics need to be added to reduce contamination from microorganisms from grains; (iii) the reaction runs at a temperature lower than the optimal temperature of the enzymes [10,12]. Therefore, this process has a lot of room for achieving a higher conversion in the future.

Finally, simple sugars convert into ethanol via yeast fermentation [34]. Yeast such as *Sacchharomyces cerevisiae* (*S. cerevisiae*) can convert glucose to ethanol and CO_2_. For each pound of simple sugars, yeast can produce approximately 0.56 L of ethanol and an equivalent amount of CO_2_ [30]. Ethanol and other by-products are separated and purified by distillation.

## 3. Starch Hydrolyzing Enzymes and Microbial Strains Used in Bioethanol Production

Starch hydrolyzing enzymes exist widely in plants, animals and microbes; thus, these enzymes or microbial strains containing starch hydrolyzing enzymes are commonly used in bioethanol production [35]. *Aspergillus* and *Rhizopus* spp. expressing α-amylase and/or glucoamylase have been used for the commercial bioethanol production [35]. Due to its high tolerance to ethanol, osmotic pressure and various inhibitors in industrial processes, *S. cerevisiae* is the preferred unicellular yeast for bioethanol production, but it lacks starch hydrolyzing enzymes for effective utilization of starch [35,36]. Therefore, many genetically modified microbial strains with efficient enzymes have been used in bioethanol fermentation processes (Table 1) [7,37,38,39,40,41,42,43].

With the help of hydrolytic enzymes (e.g., protease, pectinase and cellulase), the use of the modified strains can greatly increase the yield of ethanol and reduce the fermentation time. For example, the modified *S. cerevisiae* strain, which co-displays *Rhizopus oryzae* glucoamylase and *Streptococcus bovis* α-amylase using α-agglutinin and Flo1p (YF207/pGA11/pUFLA) [28], can produce ethanol directly from raw maize starch without addition of commercial enzymes. The strain can produce 61.8 g of ethanol /L, with 86.5% of theoretical yield from raw maize starch after 72 h of fermentation [28]. Moreover, the recombinant *S. cerevisiae* Y294[ApuA] and Y294[AteA] strains can produce high extracellular α-amylase activities, resulting in 90% reduction in the enzyme amounts required for raw starch hydrolysis [42]. By over-expression of amylase genes with strong promoters, the ability of engineering *S. cerevisiae* to convert raw starch into ethanol was significantly improved [44]. By generating transgenic maize plants overexpressing a bacterial amylopullulanase (APU) enzyme, conversion efficiency of starch into ethanol was increased to 90.5% by direct hydrolysis of the transgenic seeds using commercial amyloglucosidase [45].

Though the application of these engineering enzymes and yeast strains have successfully reduced energy consumption of fermentation, the starch hydrolysis process remains to be improved, especially on starch properties, engineering enzyme sources and reaction conditions, to convert starch into ethanol economically and efficiently.

## 4. Effect of Features of Starch on Enzymic Hydrolysis in Bioethanol Production

As discussed above, the easily degraded starchy substrates in bioethanol production would reduce energy costs and generate more economic benefits [11]. According to previous studies [46,47,48,49], the easily degraded starch usually has these characteristics: (i) low amylose content, (ii) appropriate molecular architecture and (iii) smaller size of starch granules. 

The amylose content affects the enzymatic hydrolysis rate of starch. Previously, maize starch hydrolysis experiment in vitro by human salivary α-amylase showed that the rate of digestion of 100% maize amylose < hybrid high-amylose (64–66% amylose) < waxy maize starch (99–100% amylopectin) [46]. Similar results were obtained in rice [50] and potatoes [51]. The maize resistant starch with 30% and more amylose often results in lower conversion of starch into sugars and lower final ethanol yield [52]. Amylopectin-only (waxy) maize has higher conversion efficiency of starch into ethanol than normal maize [53,54]. The conversion efficiency and ethanol yield were found to be negatively correlated with amylose content, average amylopectin branch chain lengths and the percentage of long branch chains [53,54,55,56]. High amylose starch (amylose content >70%) requires a higher gelatinization temperature and/or alkaline treatment to disrupt the hydrogen bonding [57]. Amylose is more difficult to digest than the open-branched structure of amylopectin due to the densely packed helical structure and the formation of amylose-lipid complexes [58]. Thus, it is an economical way to use an amylopectin-rich raw material that can be easily degraded to produce ethanol. 

The rate of starch enzymatic hydrolysis is controlled by starch multi-scale structures such as chain length distribution [59], crystalline structures [60], lamellar structures [61], and morphology features [62]. The amylose/amylopectin ratio and amylopectin architecture significantly affect physical and physicochemical properties of starch granules, especially gelatinization and recrystallization [47,48,63]. Starch granules with short average amylopectin branch chain lengths and with high phosphate monoester content displayed low gelatinization temperatures. Amylose influences the packing of amylopectin into crystallites and the organization of the crystalline lamellae within granules, which is important for properties related to water uptake [48]. Starch in seeds of maize *sbe1a* mutant was more resistant to digestion during germination due to the altered branching pattern of amylopectin and amylose [64]. In duckweed, the total amounts of amylose with shorter chain length negatively correlated with undigested starch content, and the amount of amylopectin long chains negatively correlated with the degradation rate [65]. Moreover, there was a report showing that starch was digested by a side-by-side mechanism, and there was no obvious preference for enzyme attack in amylopectin branch lengths, helix form, crystallinity or lamellar organization. The granule architecture was the major factor controlling enzyme susceptibility, especially the number of internal channels and pores [66]. 

The efficiency and yield of bioethanol production were also affected by starch granules sizes. Small granules of barley starch were more suitable for ethanol production than large ones, because it can produce more dextrins during enzymatic hydrolysis [49]. In addition, the higher initial rates were observed in hydrolysis of small starch granules from barley and maize, due to the higher the surface area/volume ratio of small ones [67]. The proportion of small starch granules (<8 μm) is affected by plant varieties and environmental factors. Starch granules of wheat, barley and rye show bimodal size distribution (type A, ~25 μm; type B, ~6 μm). For example, barely seed starches usually contain 5–30% small granules [49]. Starch granules of maize show unimodal size distribution, varied from 2–30 μm, with a mean size of 15.4 μm [68]. Two maize inbred lines, Zheng58 and Chang7-2 seed starches, contain about 4% and 12% small granules, respectively [69]. The formation of small and large granules involved different starch synthesis pathways and regulatory mechanisms [49,70]. However, the regulatory mechanisms of small and large granules synthesis remain to investigated.

In addition, endosperm types (e.g., floury endosperm, vitreous endosperm), starch content and other components (mainly oil, protein, mineral content) in maize and sorghum seeds affect final ethanol yield and rate of fermentation [71,72,73,74,75,76]. For example, the ratio of floury versus vitreous endosperm determines the hardness and density of the grain, thus affecting the efficiency of decortication, dry grind and wet milling processes and optimum cooking times and conditions [77]. Therefore, it is necessary to understand the process of starch biosynthesis and further to modify its structure and properties, so as to achieve efficient yeast fermentation in bioethanol production.

## 5. Starch Degradation during Maize Seed Germination

The analysis of the starch degradation pathway during seed germination can provide ideas for the optimization of starch degradation in the process of industrial fermentation. Thus, we compared the difference in starch degradation between maize seeds (in vivo) and industrial fermentation (in vitro).

Maize seeds mainly consist of endosperm and embryo, which account for 90% and 10%, respectively, of the whole dry seed weight [78]. Maize endosperm contains around 70% starch and 10% protein. The composition ratio is rather stable, because it is strictly regulated through a pre-set genetic program and affected by environmental factors [79]. 

Starch is the primary carbon and energy sources for crop seed germination and seedling early growth [80,81]. After absorbing enough water, seed germination begins at suitable temperature and pH conditions. Gibberellic acid (GA) is synthesized by the embryo and released into endosperm and aleurone layer. Then, GA induces the synthesis of hydrolytic enzymes in scutellum and aleurone cells, stimulating the mobilization of endosperm reserves and nutrients [82,83]. Finally, the endosperm solutes (e.g., soluble sugar maltodextrin, sucrose, glucose) are absorbed by scutellum and transported to the growing embryo (Figure 2A).

Starch granules in maize endosperm are generally spherical or, rarely, polygonal, surrounded by many other structures, such as protein bodies, residual walls, and amyloplast membranes (Figure 2B). The size of the starch granules varied in the range of 2–30 μm, with a mean size of 15.4 μm. Our recent study showed that the isolated starch granules exhibited typical morphological characteristics (Figure 2C) [68]. In plants, starch exists in the form of semi-crystalline, consisting of two glucose polymers: amylose and amylopectin are deposited as alternating amorphous and crystalline layers [84,85] (Figure 2D). Amylose is a linear polymer consisting of 200 to 1200 glucose units with α,1-4 glycosidic bonds. Amylopectin consists of short α,1-4 linked linear chains of 10–60 glucose units and α,1-6 linked side chains with 15–45 glucose units. An amylopectin molecule contains about 2 million glucose units [86]. The ratio of amylose and amylopectin in different varieties of maize seeds varies greatly, ranging from 0 (waxy maize) to 1 (100% maize amylose) [87]. In general, normal maize starches contain about 20–30% of amylose and 70–80% amylopectin [47,88].

Notably, the degradation of starch granules needs several amylolytic enzymes [89,90] to decompose starch granules gradually from outside to inside (Figure 2D, Table 2). The amylolytic enzymes are located in amyloplasts or extracellular space, with the maximum activities at pH 6.0–7.0 and 30 °C. α-Amylase (AMY) hydrolyses internal 1,4-α-glucosyl linkages in both amylopectin and amylose, and β-amylase (BMY) hydrolyses penultimate 1,4-α-glucosyl linkages from the non-reducing end of both amylopectin and amylose, to release the oligosaccharides. These oligosaccharides separately or in combination are known as limit dextrins and branched limit dextrins (maltotriose up to maltohexaose). Isoamylase (ISA) and limit dextrinase (LD) hydrolyse the 1,6-α-glucosyl linkages at branch points in amylopectin. Finally, malto-oligosaccharides are hydrolyzed to glucose by α-glucosidases (AGL) or glucoamylase. AGL and glucoamylase have similar functions and exist in plants, animals, bacteria and fungi [33]. The hydrolytic sensitivity of starch granules was closely related to the structure of starch granules [66]. In addition, cell wall degradation can accelerate the diffusion of amylolytic enzymes [91] and enhance the starch degradation in the endosperms.

In summary, the degradation process of starch in vivo during seed germination is a series of complicated enzymatic reactions, occurring slowly in moderate growth conditions (e.g., moisture, temperature and pH). By comparison, starch degradation in bioethanol production usually requires high temperature and violently mechanical shear to break apart starch granules from seeds and to break down starch (amylose and amylopectin) structure for easily enzyme attacks.

## 6. Molecular Proteomic and Analysis of Starch Synthesis in Maize Seeds

Starch synthesis of cereal crops is the primary determinant of grain filling [92]. The current knowledge on regulation of starch accumulation in the endosperms is mainly obtained using model cereal crops, such as rice, maize and wheat [9,18,88,92,93,94,95,96]. Identifying the key proteins/enzymes involved in starch synthesis during maize seed development is critical for understanding the molecular mechanism of starch synthesis and improving starch properties for efficient bioethanol production.

Based on the relevant studies [92,97,98,99,100], we summarized the pathway of starch biosynthesis in maize endosperms. In particular, we emphasized the difference in synthesis between amylopectin and amylose and the key enzymes that affect the ratio of amylose/amylopectin and the structure of amylopectin (Figure 3, Table 3). The entire process of starch synthesis requires the coordination of various enzymes. 

### 6.1. The Initiation of Starch Synthesis

As the substrate for primer synthesis, ADPglucose (ADPG) determines the yield of starch synthesis to a great extent. The ADPG synthesis from sucrose needs ADPglucose pyrophosphorylase (AGPase) [98] (Figure 3). AGPase was synthesized in the cytosol and then transferred into plastids. The nature and location of AGPase has remarkable variations among tissues and plant species [101]. Maize AGPase consists of one small and two large subunits (Table 3). The gene *brittle2* (*bt2*) encodes the small subunit, and the genes *shrunken2* (*sh2*) and *AGP2* encodes the large subunits. In maize endosperms, active AGPase formation requires SHRUNKEN2 and BRITTLE2 subunits [102,103]. The activity of AGPase was regulated by 3-phosphoglycerate and negatively by inorganic phosphate [88,103]. With the increase in ADPG activity in wheat endosperms, seed yield significantly increased [104]. In the mutant *sh1* (deficient in sucrose synthase), starch contents significantly decreased and soluble sugars dramatically increased [105]. The movement of cytosolic ADPG from the cytosol to the amyloplast by Brittle-1 (BT1) transporter.

Moreover, ADPG can be originated from the oxidative pentose phosphate pathway (Figure 3). Malate and triose-P are involved in oxidative pentose phosphate pathway, which provides NADPH and pentose sugars for starch synthesis. Two starch phosphorylases (Pho1 and Pho2), located in the amyloplast and cytosol, respectively, were found to play a critical role in this process [106]. In rice, Pho1 is relatively specific in endosperm and related to the initiation of starch synthesis during seed development [107].

### 6.2. Amylose Synthesis

In amyloplast, the amylose biosynthesis from ADPG is mainly catalyzed by granule-bound starch synthase (GBSS) (encoded by *Wx*) (Figure 3). Several GBSS and GBSSI have been identified in maize endosperms (Table 3). Compared with starch synthase (SS), GBSSI has a lower affinity for ADPG, so it is more suitable to catalyze the synthesis of amylose at a high concentration of ADPG [108]. The decrease in or loss of GBSS enzyme activity not only reduce the amylose content, but also significantly reduce the long amylopectin content in amylopectin [109]. Twenty-seven loci linked to amylose content were identified by genome-wide association analysis in 464 inbred maize lines. These loci contain 39 candidate genes, e.g., transcription factors, glycosyltransferases, glycosidases, hydrolases [110].

### 6.3. Amylopectin Synthesis

For amylopectin synthesis in amyloplast, transglycosylating branching enzyme (BE) is the only enzyme that forms 1,6-α-branch points in amylopectin molecules. Two classes of BE exists in cereal crops: BEI and BEII (two isoforms: BEIIa and BEIIb). In the mutants (*amylose-extender*, *ae*) of maize and rice, amylopectin structure was found to be shaped by BEIIb [111,112,113,114]. BEIIb synthesized short chains with a 6–13 degree of polymerization (DP), starch branching enzyme I (SBEI) synthesized intermediate chains DP11-22, and SBEIIa also synthesized intermediate chains in rice endosperm [115,116]. In the maize mutants deficient in BEIIb [50,104] and starch synthase IIa (SSIIa) [117], a large amount of amylose (>85%) accumulated in endosperm starch.

In the mutant of *isa1*, the structure of amylopectin showed a dramatic change, and the contents of total starch, amylose and amylopectin were significantly reduced in rice endosperm [118]. The debranching enzyme isoamylase (ISA), limit dextrinase (LD) and pullulanase (PLA) catalyzes the hydrolysis of 1,6-branch points in amylopectin, which affects the internal chain length and the cluster structure of amylopectin [119,120]. In maize loss-function mutant *sugary1* (*su1*), the structure of amylopectin was changed with the increased short glucans [121]. Moreover, the increased number of small granules was observed in endosperm of the barely mutant lacking ISA1 activity [122]. Maize PLA partially compensates for the defect in ISA in the zpu1-204 endosperm [119]. SBEIIb and ISAII are negatively correlated with the contents of amylose and long amylopectin chains (DP > 30) and positively correlated with the content of short amylopectin chains (DP ≤ 31) and the molecular size of amylopectin molecules [123]. 

In plants, there are five classes of SS. These SSs are similar at the C-terminal region, but obviously differed at the N-terminal region, which provide unique functions. SSI, SSIIa and SSIIIa are also required for the synthesis of amylopectin in maize. They elongate amylopectin chains with DP 6–7 to DP 8–12, DP 6–12 to DP 13–24, and long chains connecting amylopectin clusters, respectively [88,124]. SSI is entrapped as a relatively inactive protein within the starch granule and glucan extension for continuation of amylopectin synthesis requires other SS enzymes [125]. The *opaque2* in rice can change the amylopectin branching patterns [126]. 

### 6.4. Key Enzymes, Proteins, Transcription Factors and Other Regulatory Factors Involved in Starch Synthesis

Proteome is the entire set of proteins expressed by a genome under particular conditions, and it represents the key enzymes and proteins in biochemical processes [25]. Proteomics is a large-scale approach that is widely used to catalog and identify proteins in biochemical processes at different proteome states or environments [26]. A more complete understanding of the entire process of starch synthesis is an essential prerequisite for manipulation of the process. To date, proteomic analyses have identified a lot of proteins/enzymes involved in starch synthesis in seeds of maize [97,127,128,129] and rice [130,131], wheat [132]. These proteins/enzymes can be classified as ADPG (e.g., UDP-glucosyltransferase; glucose-1-phosphate adenylyltransferase; AGPase), amyloses (e.g., GBSS) and those related to amylopectin synthesis (e.g., SS, SBEIIa, SBEIIb) (Table 3). Many of them exist in multiple isoforms.

In maize endosperms, certain SSs and SBEs exist in multisubunit complexes (SSI/SSIIa, SSI/SSIII, SSI/BEI, SSI/BEIIa, SSI/BEIIb, SSIIa/BEIIa, SSIIa/BEIIb) [92,99,133]. In particular, SBEI cannot function without SBEIIa or SBEIIb [134]. In maize, the loss of SBEIIa activity was detected in both *pul* and *isa* mutants [94,121]. Thus, it is suggested that the coordinated regulation of SS, SBE and DBE enzyme activities in starch synthesis pathway is likely to be completed in the form of complex. Phosphorylation is necessary for the formation of the complex of SS, SBE and DBE. After dephosphorylation, SBEI, SBEIIb and starch phosphorylase could no longer form a complex in vitro [135]. SSIIa, SBEIIa and SBEIIb will form a 300 kDa complex, whereas SSIIa, SSIII, SBEIIa and SBEIIb form a 670 kDa complex in wild-type maize [99]. However, the specific roles of these enzymes and means of coordination remain to be characterized.

Some non-enzyme proteins are involved in starch synthesis. 14-3-3 protein, one of regulatory factors family, was associated with BEII, SSI, and SSII in the developing barely endosperm [136]. Protein targeting to starch (PTST) was found to be necessary for starch synthesis in Arabidopsis [137]. Additionally, the activities of enzymes related to starch synthesis were regulated at transcription level [9,88]. The promoter region of *sbeIIb* contains sucrose response elements (SURE), which can regulate seed starch synthesis based on sugar availability. The details in the regulatory network of starch synthesis needs to be clarified in the future. 

Moreover, transcription factors are involved in controlling starch biosynthesis, starch content and amylose/amylopectin ratio, such as ZmMADS1a, ZmCBM48-1, ZmbZIP22 and ZmMYB14 [138,139,140,141]. In addition to the down-regulation and overexpression of one or more genes, starch properties can also be improved through post-translational modification sites or regulatory enzymes [142]. For example, the change in phosphorylation at Ser-34 position affects the activity of GBSSI, which leads to the decrease in amylose content in wheat [143].

In summary, the knowledge on starch biosynthesis in maize endosperm, especially the identification of key enzymes, proteins and regulatory factors, can provide the clue to change the properties of starch and to redesign special starch for bioethanol production [144,145,146]. However, the regulation of the ratio of amylose/amylopectin and their total amounts involves a set of fine regulation mechanisms, which requires further research.

## 7. Genetic Engineering of Starch Biosynthesis in Maize Seeds for Bioethanol Production

After long term domestication and cultivation, modern maize has a large number of phenotypic and genotype diversities. Common maize is classified as flint, pop, flour, dent, sweet maize, etc., mainly based on the morphological, rheological, functional and thermal properties of starch in endosperms [147]. For example, dents maize seeds have indented characteristic and high soft starch content, whereas flints maize seeds have a thick, hard, and vitreous outer layer [148]. Most commercial maize used in bioethanol production is dent type, especially the dent maize hybrids due to its high yield and stress tolerance. However, some dent maize hybrids are unsuitable for industrial processing because they contain too little vitreous endosperm. These starch properties were affected by hereditary and environmental factors. 

A combinatorial approach through genomics, proteomics, and other associated-omics branches of biotechnology is proving to be an effective way for accelerating genes or enzymes discovery and the crop improvement programs. Compared with traditional plant breeding methods (e.g., cross-breeding, mutation breeding), genetic engineering techniques can precisely target DNA sequence to obtain desired agronomic traits (e.g., high quality, high yield and stress resistance) of plants [18,19,20,21,22]. Especially, CRISPR (clustered regularly inter-spaced short palindromic repeats)-Cas (CRISPR associated), as one of the most advanced technologies for engineering crop genomes, has been rapidly expanding and applied to major cereals such as rice, wheat and maize in recent years [22]. 

Below, we have summarized recent studies on modulation of starch composition, structure and properties in cereal crops through genetic engineering techniques (Table 4) [23,24,149,150,151,152,153,154,155,156,157,158]. For example, a high-amylose maize line (amylose content > 50%) was obtained using RNAi in *SBEIIa* or *SBEIIb*, with better applications in the areas of films, foods, medical treatments and textiles [23,24]. By simultaneous overexpression of *Bt2*, *Sh2*, *Sh1* and *GBSSIIa* and suppression of *SBEI* and *SBEIIb* by RNAi, the starch content increased 2.8–7.7% and amylose content increased 37.8–43.7% in maize endosperm. Additionally, the 100-grain weight increased 20.1–34.7% [158]. DuPont Pioneer knocked out the maize waxy gene *Wx1*, which encodes the GBSS gene that is responsible for making amylose. In the absence of GBSS expression in the endosperm, amylose was not synthesized, and this created a high amylopectin (waxy) maize with improved digestibility and the potential for bio-industrial applications [20]. The limitations of *Wx* mutant allele acquisition and breeding efficiency by conversion of parental lines from normal to waxy maize were overcame [154].

RNAi construct may silence multiple genes if these genes have high similarity. For example, *TaSBEIIa* and *TaSBEIIb* of wheat were silenced simultaneously by RNAi [159]. Many enzymes involved in starch metabolism have multiple isoforms. Different isoforms may have different functions (e.g., BEI, BEIIa and BEIIb) [134]. Therefore, CRISPR/Cas9, as a simple, versatile, robust and cost-effective system, will be a useful tool for generation of sequence-specific targeted mutagenesis for maize easily degraded starch genome manipulation and bioethanol production improvement.

CRISPR/Cas9 system can be used to target the genes/enzymes related to starch biosynthesis pathways [156,160]. It will help in the improvement of structure and physicochemical properties of starch (Table 4). By knocking out the *Wx* gene from two elite cultivated rice lines (XS134 and 9522) with CRISPR/Cas9 technology, glutinous (sticky) rice varieties with reduced amylose content (0–5%) were developed, resulting in better brewing performance [155]. A similar study was performed in rice inbred lines Huaidao 5 and Suken 118 [160]. Targeted mutagenesis in *SBEI* and *SBEIIb* in rice were obtained using CRISPR/Cas9 technology. The content of amylose and resistant starch significantly increased in the *sbeIIb* mutant, but no difference in starch features was observed between the *sbeI* mutant and its wild type [161]. After knocking out the genes *IbGBSSI* (encoding granule-bound starch synthase I) and *IbSBEII* (encoding starch branching enzyme II), the amylose percentage in sweet potato (*ipomoea batatas*) starch reached 5.5% and 40.3%, respectively. The two potatoes with different starch compositions are excellent parental lines in genetic crossing to produce novel properties for food and industrial applications [156]. The gene *Wx1* has been targeted in the potato to produce waxy potatoes, with improved cultivars aimed predominantly at the industrial starch market [162]. Therefore, targeting biosynthetic pathway genes of starch can effectively alter starch functionality. 

Compared to many allopolyploid crops, such as wheat, potato, and *Brassica napus*, the genomic study of the maize is relatively simple. Similar results may be obtained by manipulating homologous target genes in maize by analyzing the role of some target genes in improving starch in other crops. Therefore, the design of transgenic maize by genetic engineering strategies will produce unique starches as raw materials for bioethanol production [163]. It is feasible that elite maize hybrids with optimized starch feature, high yield and stress tolerance can be developed by crossing of genetically modified inbred lines, which have a low energy cost during the bioethanol production.

We are still facing many challenges in creating easily degraded maize starch as feedstock in bioethanol industry. Although the pathway of starch metabolism is largely clear, the network and molecular mechanism of regulation involved in starch biosynthesis remains unknown, especially the regulation of the ratio of amylose/amylopectin and their accumulation. Many genes and regulatory factors have been found to contribute to starch biosynthesis, but the key genes are still unclear. Thus, we here summarized proteomic and molecular studies on maize starch metabolism and indicated the key genes of amylose and amylopectin. Alterations in starch structure can be achieved by modifying genes encoding the enzymes responsible for starch synthesis. Based on the discussion above, it is proposed that GBSS, BE and DBE among others (Table 3) are the most promising target of genetic engineering for molecular breeding to reduce the synthesis of amylose, change the structure of amylopectin or increase the number of small granules in maize amyloplasts. The studies with transgenic lines show that with decreased abundance or enzymatic activities of specific SS or increased abundance or enzymatic activities of BE or DBE, separately or simultaneously, we can produce starch granules with decreased crystallinity, thus increasing the susceptibility to efficient enzymatic digestion for improved bioethanol production.

## 8. Conclusions

At present, maize kernels are the main raw material used for bioethanol production. However, the composition of common maize endosperm starch is far from optimum to serve as feedstocks for yeast fermentation in bioethanol production. In particular, the amylose/amylopectin ratio, amylopectin architecture and the number of small starch granules need to be modified by genome editing technology for efficient enzymatic digestion during bioethanol production. The discovery and isolation of target genes is the premise of genetic engineering. Proteomic analysis and transgenic studies have identified the key enzymes and regulatory factors involved in starch biosynthesis. The knowledge on controlling the synthesis of amylose to amylopectin and the formation of starch structures provides a clue for designing easily degraded maize starch as feedstock in the bioethanol industry.

## Figures and Tables

**Figure 1 ijms-24-03927-f001:**
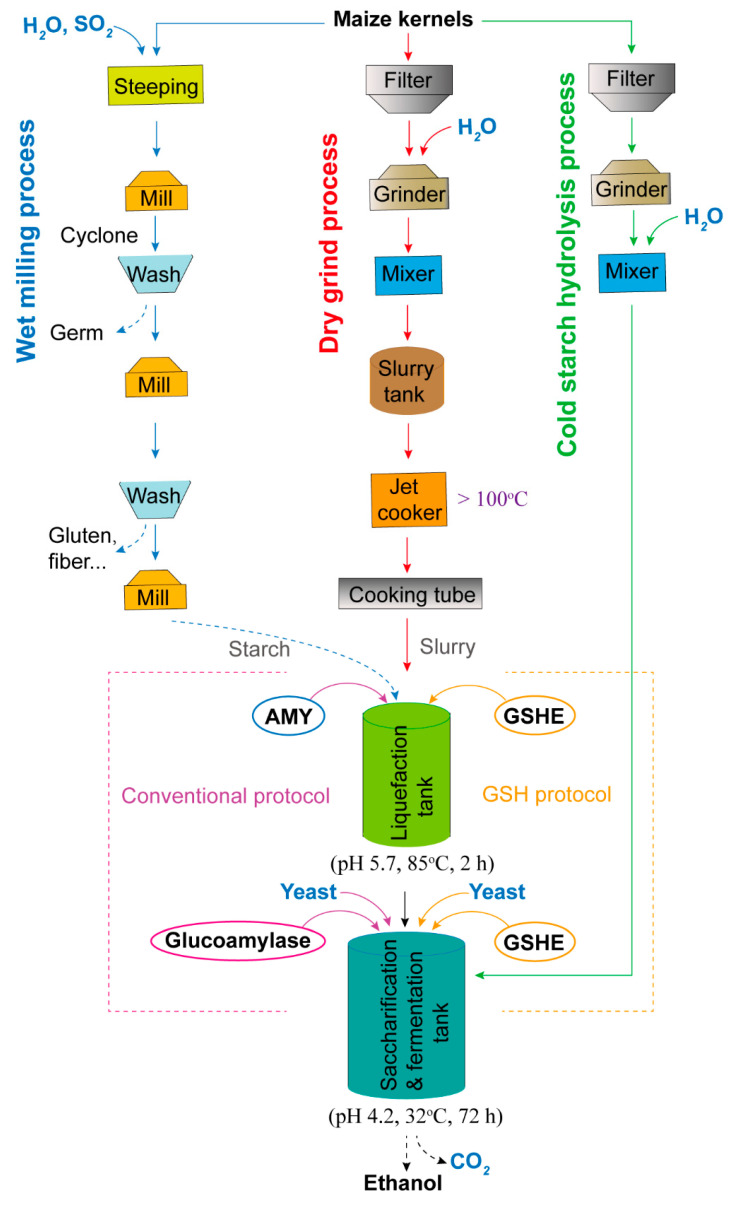
Flow chart of the wet milling process, dry grind process and cold starch hydrolysis process for bioethanol production. Left panel, conventional protocol; right panel, GSH protocol. AMY, α-amylase; GSH, granular starch hydrolysis; GSHE, granular starch hydrolysis enzymes.

**Figure 2 ijms-24-03927-f002:**
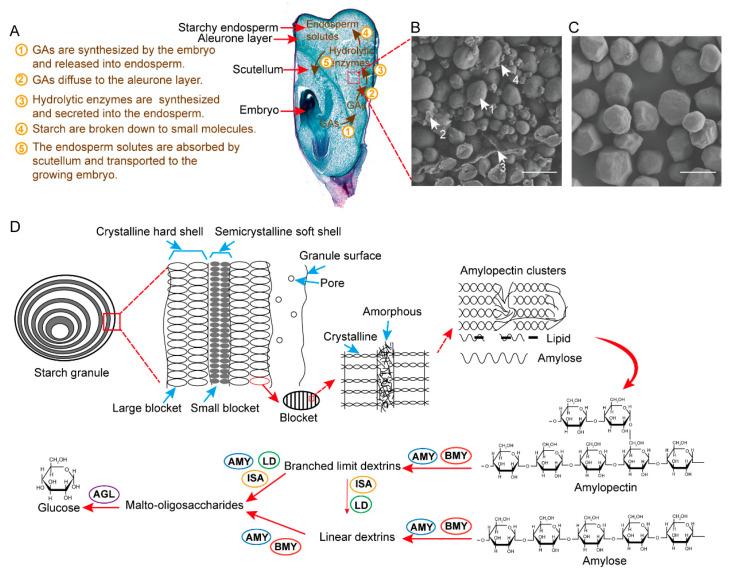
Starch granules in maize kernel and starch degradation during seed germination. The key enzymes involved in starch degradation in vivo were indicated. (**A**) Starch degradation represented within a seed. GA, gibberellic acid. (**B**) Scanning electron microscopy of starch granules in maize endosperm [68]. Bar = 20 μm. 1, starch granule; 2, protein body; 3, cell wall; 4, amorphous debris. (**C**) Scanning electron microscopy of isolated starch granules [68]. Bar = 20 μm. (**D**) Starch degradation pathway in vivo. AMY, α-amylase; BMY, β-amylase; ISA, isoamylase; LD, limit dextrinase; AGL, α-glucosidase.

**Figure 3 ijms-24-03927-f003:**
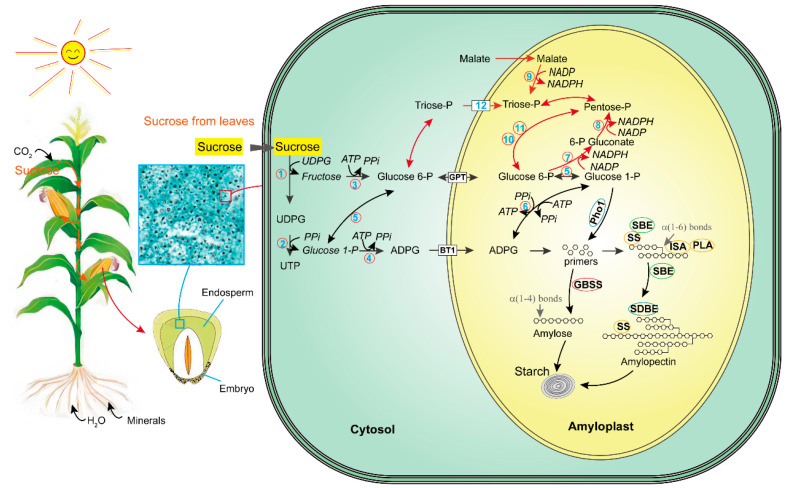
Starch biosynthesis in maize endosperm. Black arrows show the conventional biosynthesis of starch: glucose-1-P is converted to ADPG, which is transported into the amyloplast and polymerized into starch. Red arrows show the oxidative pentose phosphate pathway, which provides NADPH and pentose sugars for starch synthesis. 1, sucrose synthase; 2, UDP-glucose pyrophosphorylase; 3, hexokinase; 4, AGPase; 5, phosphoglucomutase; 6, complexes including AGPase, starch synthase IIa (SSIIa), SSIII, starch branching enzyme IIb (SBEIIb), and SBEIIa; 7, glucose-6-phosphate dehydrogenase; 8, 6-phosphogluconate dehydrogenase; 9, malic enzyme; 10, transaldolase; 11, transketolase; 12, triose-P/P translocator and P/phosphoenolpyruvate translocator.

**Table 1 ijms-24-03927-t001:** Microbial strains, enzymes used in laboratory and industrial ethanol fermentation.

Starch Sources	Microbial Strains	Enzymes/Additives	Reaction Conditions	Reaction Volume	Ethanol Yield	Reference
Raw maize starch	*S. cerevisiae* ATCC 9763 distiller’s yeast expressing both GA1 and AMY	_	30 °C	100 mL	80.9 g/L	[37]
Sweet potato tuberous roots	*Zymomonas mobilis* ZM4	_	30 °C, pH4.0, 72 h	250 mL	14.4 g/100 g sweet potato	[38]
Raw maize starch	*Kluyveromyces marxianus* YRL 009	_	42 °C	250 mL	56.82 g/L	[39]
Maize, soybean	Bacto™ yeast extract; *Escherichia coli* KO11	Fiber-hydrolyzing enzymes; soy skim; insoluble fiber	α-Amylase, 85 °C, 3 h; glucoamylase, dry yeast, 30 °C, pH 4.5, 68 h	250 mL	3.20 g/h/100 g dry maize	[40]
Raw maize and cassava flours	*Penicillium* sp. *GXU20*; a thermo-resistant dried *S. cerevisiae*	α-Amylase; starch-digesting glucoamylase	pH 4.0, 40 °C, 48 h	50 mL	61.0 g/L	[7]
Maize	*S. cerevisiae*	Fermgen (a protease); pectinase and cellulase; Tween^®^ 80	α-Amylase, 81 °C, 3 h; glucoamylase, dry yeast, 30 °C, pH 4.0, 64 h	250 mL	34.98 g/100 g dry maize	[41]
Raw maize starch	*S. cerevisiae* Y294	Glucoamylase plus STARGEN 002 amylase cocktail™	30 °C	120 mL	73.80 g/L	[42]
Raw maize starch	*S. cerevisiae* ER T12 and M2n T1	STARGEN 002 amylase cocktail™	30 °C	250 or 500 mL	89.35 and 98.13 g/L	[43]

**Table 2 ijms-24-03927-t002:** Major enzymes identified in the starch degradation during maize seed germination.

Name	Other Name	Accession	Molecular Function	Product	Reaction Conditions	Location
α-Amylase (EC:3.2.1.1)	1,4-α-Glucan endohydrolase	B4G231	α-Amylase activity, breaking down internal 1,4-α-glucosyl bonds	Maltose, maltotriose, oligosaccharides, limit dextrins	pH 6.7–7.0	Extracellular space
Isoamylase (EC:3.2.1.68)	1,6-α-Glucan endohydrolase	A0A3L6EGI3, A0A3L6EH01	Amylase activity, breaking down internal 1,6-α-glucosyl bonds	Maltodecaose	pH 6.5, 30 °C	Plastid
β-Amylase (EC:3.2.1.2)	1,4-α-Glucan exohydrolase	P55005	β-Amylase activity, breaking down internal 1,4-α-glucosyl bonds	Maltodecaose	pH 6.0–7.0	Plastid
Limit dextrinase (EC:3.2.1.142)	Pullulanase	O81638	Pullulanase activity, breaking down remaining 1,6-α-glucosyl bonds	Maltodecaose, dextrins	pH 6.5, 30 °C	Plastid
Glucoamylase (EC:3.2.1.3)	γ-Amylase	_	γ-Amylase activity, breaking down terminal 1,4-α-glucosyl bonds to produce β-D-glucose	Glucose	_	_
α-Glucosidase (EC:3.2.1.20)	Maltase	A0A1D6HXT2, A0A1D6EMY1	α-Glucosidase activity, breaking down terminal1,4-α-glucosyl bonds to produce α-D-glucose	Glucose	_	Extracellular space

**Table 3 ijms-24-03927-t003:** Key enzymes identified by proteomic analyses of starch biosynthesis during maize seed development.

Enzyme Name (EC Number)	UniProt Accession	Coding Gene	Subcellular Localization	Molecular Function
Sucrose synthase (EC:2.4.1.13)	D2IQA1	*sh1*	Plastid, cytosol, nucleus *	Sucrose synthase activity
Sucrose synthase 2 (EC:2.4.1.13)	P49036	*SUS1*	Amyloplast *	Sucrose synthase activity
UDP-glucosyltransferase	C0LNQ9	*N/A*	Amyloplast *	UDP-glycosyltransferase activity
UTP-glucose-1-phosphate uridylyltransferase (EC:2.7.7.9)	B4FAD9	*100191846*	Cytosol *	Regulating UDP glucose pyrophosphorylase activity
Glucose-1-phosphate adenylyltransferase (large subunit 1) (EC:2.7.7.27)	P55241	*SH2*	Amyloplast	Involvement of ADP-glucose synthesis
AGPase (EC:2.7.7.27)	Q947B9, A5GZ73	*542295*, *100101531*	Amyloplast	Involvement of ADP-glucose synthes
AGPase small subunit (EC:2.7.7.27)	Q941P2, D3YKV1	542072	Amyloplast	Involvement of ADP-glucose synthesis
AGPase large subunit 1 (EC:2.7.7.27)	P55241	*SH2*	Amyloplast	Involvement of ADP-glucose synthesis
AGPase large subunit 2 (EC:2.7.7.27)	P55234	*AGP2*	Amyloplast	Involvement of ADP-glucose synthesis
Fructokinase-1 (EC:2.7.1.4)	Q6XZ79	*FRK1*	Cytosol	Fructokinase activity
Phosphoglucomutase (EC:5.4.2.2)	P93805	*N/A*	Cytosol	Interconverting glucose-6-P and glucose-1-P
α-1,4-glucan phosphorylase (EC:2.4.1.1)	B5AMJ8	*100285259*	Amyloplast *	Glycogen phosphorylase activity
Granule-bound starch synthase	Q93WP1, Q94FZ6	*waxy*	Amyloplast *	Transferring glycosyl groups
Granule-bound starch synthase 1	A0A1D6L3I4, P04713, I1VEV9	*541854*, *WAXY*, *wx1*	Amyloplast	Transferring glycosyl groups
Starch synthase	A0A1D6LVT4, O49064, O48899	*SS1*, *zSSIIa*, *sh1*	Amyloplast	Starch synthase activity
Amylose extender starch-branching enzyme	Q84QF8	*ae1*	Amyloplast *	Extending 1,4-α-glucosyl linkages
Starch branching enzyme IIa (EC:2.4.1.18)	A0A1D6EBS5	*542342*	Amyloplast	Extending 1,4-α-glucosyl linkages
Starch branching enzyme IIb (EC:2.4.1.18)	Q7XZL1	*ae1*	Amyloplast *	Extending 1,4-α-glucosyl linkages
Starch branching enzyme III (EC:2.4.1.18)	K7VJE7	*Zm00001d011301*	Amyloplast	Extending 1,4-α-glucosyl linkages
1,4-α-Glucan-branching enzyme (EC:2.4.1.18)	A0A1D6H9R5	*Zm00001d016684*	Amyloplast	Extending 1,4-α-glucosyl linkages
Pullulanase-type starch debranching enzyme	A0A0H4FPZ7	*Zpu1*	Amyloplast *	Pullulanase activity
Isoamylase-type starch debranching enzyme3	A0A1D6I6A1	*Zm00001d020799*	Amyloplast *	Hydrolase activity
SU1 isoamylase (EC:3.2.1.68)	O22637	*sugary1*	Amyloplast *	Hydrolase activity
Amylomaltase (EC:2.4.1.25)	A0A1D6INP5	*Zm00001d022510*	Amyloplast *	4-α-Glucanotransferase activity

Note: * indicates subcellular localization information that was predicted using Plant-mPLoc (http://www.csbio.sjtu.edu.cn/bioinf/plant-multi/; accessed on 10 June 2021).

**Table 4 ijms-24-03927-t004:** Modulation of starch composition, structure and properties in cereal crops through genetic engineering techniques.

Plant Species	Target Genes	Loss or Gain-of-Function	Molecular Tools Used	Main Changes in Starch Properties and Other Phenotype in Mutant Lines	Reference
Winter wheat (cv Zhengmai 7698); spring wheat (cv Bobwhite)	*SBEIIa*	Knock-out	CRISPR–Cas9	The total starch contents decreased slightly, and the amylose contents increased significantly. The shape of starch granules was highly irregular. The percentage of A-type and B-type starch granules increased, and C-type starch granules decreased. The proportion of shorter-chain amylopectin with degree of polymerization (DP) of 6–8 and larger chains >18 DP significantly increased, and the proportion of DP 9–17 decreased. The number of spikelet and grain number per main spike, the length and width of grains, and the 1000-grain weight decreased.	[149]
*Brassica napus*	*SBEs*	Knock-out	CRISPR–Cas9	The pattern of amylopectin chain length distribution was altered. The shape of starch granules was highly irregular. Starch synthesis and turnover became slow in the leaves. There was no significant difference in the rate of oil biosynthesis, final oil content, and fatty acid composition.	[150]
Potato (*Solanum tuberosum* L.)	*SBE3*	Knock-out	CRISPR/dMac3-Cas9	The amylose content was significantly reduced. The mutant lines grew normally and showed a similar morphology to that of the wild type. The tubers shaped normally were similar to the wild-type plant.	[151]
Potato (*Solanum tuberosum* L.) Yukon Gold strain TXYG79	*GBSSI*	Knock-out	CRISPR–Cas9	Only 4.4% amylose was detected in the tubers. A significantly higher peak and final viscosity was observed. No obvious differences in yield or morphology (skin color, shape and size) of the tubers were observed.	[152]
Rice (*Oryza sativa* ssp. *Japonica* cv. EYI)	*GBSSI*	Knock-out	CRISPR–Cas9	GBSS activity in the mutants was 61–71% that of wild-type levels. The amylose content declined to 8–12% in heterozygous seeds and 5% in homozygous seeds. The aleurone layer and amorphous starch grain structures were abnormal.	[153]
Maize inbred line H99	*SBEIIa*	Loss-of-function mutation	RNAi	SBE activity was decreased by up to 67.8%. Total starch content had no significant difference, but the percentage of amylose was increased to approximately 66.8% versus the control.	[23]
Maize inbred line Chang7-2	*SBEIIa, SBEIIb*	Loss-of-function mutation	RNAi	SBE activity decreased to 40% that of the wild type. Starch content of kernels decreased slightly (from 66% to 63%). The amylose content was greatly increased to 41.86–55.89%. There were more long chains in amylopectin. The shape of starch granules was irregular. Similar plant phenotypes and kernel production were observed.	[24]
Maize	*Waxy*	Knock-out	CRISPR–Cas9	The average amylopectin content was 94.9%. The plant height, ear height, and grain yield, kernel row number, kernel per row and hundred-kernel weight were not significantly different.	[154]
Rice (*Oryza sativa* ssp. *Japonica* cv. XS134)	*Waxy*	Knock-out	CRISPR–Cas9	Total starch content was unchanged. Amylose content was significantly reduced. Starch shape was more irregular. No differences in plant height, grain number per panicle, panicle number per plant, yield per plot, seed width, seed length and 1000-grain weight were observed.	[155]
Sweet potato (*Ipomoea batatas*)	*GBSSI, SBEII*	Knock-out	CRISPR–Cas9	Total starch content was not significantly changed. The amylose content was reduced in the *IbGBSSI*-knockout mutant and increased in the *IbSBEII*-knockout mutant. There were fewer short chains and more long chains in *IbSBEII*-knockout mutant.	[156]
Potato (*Solanum tuberosum* L. Cv. Solara)	*ISA1, ISA2 and ISA3*	Loss-of-function mutation	RNAi	Total starch content and the size of starch granules decreased. There was no significant change in chain length composition. The number of small starch granules increased.	[157]
Maize inbred line H99	*Bt2, Sh2, Sh1, GbssIIa, SbeI, SbeIIb*	Loss or gain-of-function	RNAi, overexpression	Starch content increased 2.8–7.7%, and the amylose content increased 37.8–43.7%. The 100-grain weight increased 20.1–34.7%, and the ear weight increased 13.9–19.0%.	[158]

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
