# Peer review of "Genetic Engineering of Starch Biosynthesis in Maize Seeds for Efficient Enzymatic Digestion of Starch during Bioethanol Production"

_ijms, 2023, doi:10.3390/ijms24043927_

Round 1

Reviewer 1 Report

This paper reviews the present status of starch biosynthesis in maize seeds for bioethanol production. The article is systematic, allowing readers to clearly understand the latest research progress in related fields. Some problems are as following:

1. In section 1, authors should explain why those genetic technologies to produce easily-degraded starch for bioethanol production are seldom used in maize.

2. In line 107, the meaning of GSH should be pointed when it is present at first time. Same for “SS” in line 348.

3. In line 218-219, there is something wrong with the grammar of this sentence of ... Thus, we below compared the difference of maize seeds starch degradation in vivo and in vitro…”. Many sentences should be grammatically modified. In line 341, “…In maize mutant sugary1 (su1) which loss-function of ISA1…”;

4. In line 256, “30oC” should be “30 ℃”.

5. What is the difficulty to breeding easily-degraded maize starch as feedstock in the bioethanol industry? Authors should give a clear statement.

Author Response

We are grateful to you for taking time to review the manuscript and provide us with valuable comments. The submission has been substantially revised after incorporating all the comments. All the grammar and writing mistakes indicated have been corrected. In addition, we have carefully checked the format of our manuscript according to the submission guidelines of the International Journal of Molecular Sciences. The changes made in the revision are highlighted in blue. Below are our point-to-point responses to the comments raised by the referees.

  1. Thanks to your comments. We have added relevant content in Section 1 (lines 77-88). As one of the main crops, maize is usually used as food for people and animals as well as industrial raw materials. At present, traditional breeding and molecular breeding are mainly aimed at obtaining crops with higher quality, high yield and high stress resistance. Although some breeders have modified maize starch to better meet the food and industrial needs, there are no maize varieties specifically used for bioethanol production. Also, the starch biosynthesis in maize endosperms is a complex process that requires the coordination of various enzymes and regulatory factors. Thus, inadequate understanding of the regulation pathway of starch synthesis and potential key target genes led to less use of genetic technology to improve maize starch for bioethanol production. With the generation of advanced high-throughput genomic, transcriptomic, proteomic data, our understanding of the key enzymes and regulators of starch synthesis regulation pathway will be more profound, which will provide new ideas for the future use of genetic technologies to produce easily-degraded starch for bioethanol production.
  2. We have added the definition information of GSH (line 119) and SS (line 332). In addition, the definition of abbreviations that first appeared in the full text is checked and supplemented in the revised manuscript. 
  3.  The grammar and writing mistakes have been corrected (lines 230-232, lines 352-353).

  4. The mistake has been corrected in the revised manuscript (line 268).
  5. We added relevant content in the Section 7 (Genetic engineering of starch biosynthesis in maize seeds for bioethanol production) (lines 473-479). We are still facing many challenges in creating easily-degraded maize starch as feedstock in bioethanol industry. Although the pathway of starch metabolism is largely clear, the network and molecular mechanism of regulation involved in starch biosynthesis remains unknown, especially the regulation of the ratio of amylose/amylopectin and their accumulation. Many genes and regulatory factors have been found to contribute to starch biosynthesis, but the key genes are still unclear.

Reviewer 2 Report

Dear authors

First of all, let me congratulate you for a such interesting paper. You've done a very good job with the important amount of bibliography used and in the summarizing process.

The description of the process of fermentation and the relationships with enzymes is coherent and correct. As well, the information described in Tables is, totally actualized and correctly organized.

After several times reading your work, I have only detected two little mistakes in format. It would be very sad, to see them in the final version.

Line 66. Please, consider revising the format of the Tons of cereal seeds (at the beginning of the phrase)

Line 428. The format in which wx is written, can give to confusion. Please, consider rewriting as Wx

Many thanks

Author Response

We are grateful to you for taking time to review the manuscript and provide us with valuable comments. The submission has been substantially revised after incorporating all the comments. All the grammar and writing mistakes indicated have been corrected. In addition, we have carefully checked the format of our manuscript according to the submission guidelines of the International Journal of Molecular Sciences. The changes made in the revision are highlighted in blue.

Thank you very much again for your encouragement and support of our work. The mistakes in format have been corrected in the revised manuscript (line 67, line 440).

Reviewer 3 Report

The authors try to present in their manuscript a review regarding some elements of bioengineering regarding bioethanol obtained from maize starch, i.e the advances made in molecular characterization of the key proteins involved in starch metabolism in maize seeds. The review explores how these proteins affect the starch metabolism pathway, especially in controlling the composition, size and features of starch. The authors try to highlight the roles of key enzymes in controlling the amylose/amylopectin ratio. Based on the current technologies and processes of bioethanol production using maize starch, the authors propose the modifications of several key enzymes  (as abundance or activities, by genetic engineering) to synthesize easily-degraded starch granules in maize seeds.                       From my point of view, the subject of this article is not suitable for the International Journal of Molecular Sciences, that because the manuscript contains more elements from engineering and biotechnology. Additionally,  the information from the manuscript is not presented in an original manner for a manuscript type  Review. From my point of view, this article is more suitable for other MDPI Journals such as  Applied Sciences or BioTec.

Author Response

We are grateful to you for taking time to review the manuscript and provide us with valuable comments. The submission has been substantially revised after incorporating all the comments. All the grammar and writing mistakes indicated have been corrected. In addition, we have carefully checked the format of our manuscript according to the submission guidelines of the International Journal of Molecular Sciences. The changes made in the revision are highlighted in blue. Below are our point-to-point responses to the comments raised by the referees.

Thank you very much for your comments. This review was submitted at the invitation of the editors. This paper summarizes the key processes of bioethanol production from maize starch, and the relevant contents represent a small part of the full text (Sections 2 and 3), focusing on the properties of starch and their effects on bioethanol production. In Sections 4, 5, and 6, this review analyzed the physical and chemical properties of easily degradable starch, the difference between the germination process and the degradation in vitro of maize seeds, and analyzed the starch metabolism pathway of maize seeds using molecular proteomics, especially discussed how key enzymes control the composition, size, and characteristics of starch. The potential key enzyme proposed by us can be used as the potential targets of genetic engineering to improve starch properties for bioethanol production. In our opinion, this paper may fit the topic of the special issue "Advances in Revealing Starch Molecular Structure, Functionalities and Biosynthesis".

Round 2

Reviewer 3 Report

The articles sent to the International Journal of Molecular Sciences must belong to the field of molecular biology and molecular medicine, with the accent on the last one. The authors present a review in the field of the technology applied sciences. I read the article with attention again and my opinion remains the same:  the problems treated in this manuscript are not proper with the journal's aims. The more proper journals for this manuscript are Processes, Applied Sciences, Fermentation, Applied Sciences, Bioengineering or BioTech.